# A RATE–DISTORTION VIEW ON MODEL UPDATES

**Nicole Mitchell, Johannes Ballé, Zachary Charles & Jakub Konečný**
Google Research
{nicolemitchell, jballe, zachcharles}@google.com

## ABSTRACT

Compressing model updates is critical for reducing communication costs in federated learning. We examine the problem using rate–distortion theory to present a compression method that is near-optimal in many use cases. We empirically show that common transforms applied to model updates in standard compression algorithms, normalization in QSGD and random rotation in DRIVE, yield suboptimal compressed representations in practice.

## 1 INTRODUCTION

Federated learning (FL) is a machine learning framework in which clients collaboratively train a model under the coordination of a central server, without sharing their local data. Prototypical FL algorithms such as FedAvg (McMahan et al., 2017) involve multiple communication rounds interleaving local training and model update aggregation. Network constraints often make communication a bottleneck in FL training (Kairouz et al., 2021).

This work focuses on compressing model updates to improve communication efficiency, which is closely related to gradient compression for distributed training (DT). The analysis and effectiveness of existing FL and DT compression methods is often based on worst-case guarantees (Alistarh et al., 2017; Vargaftik et al., 2021). By exploiting the consistent structure of model updates in FL and leveraging rate–distortion optimization to guide the design and usage of our method, we generate a more efficient representation for the *average* case on the pareto frontier of compressed size (*bitrate R*) and fidelity (*distortion D*).

## 2 METHOD

An FL compression method consists of a pair of operators $(\mathcal{E}, \mathcal{D})$ to encode each client's model update and decode each compressed packet on the server (Appendix A.1). Our design of $\mathcal{E}$, given in Algorithm 1, is informed by the observation that the coordinates of weighted model updates follow a consistent symmetric, unimodal, sparse, and heavy-tailed distribution centered around zero (A.3).

---

**Algorithm 1** Client-side encoder $\mathcal{E}$

**Let:** $\square$ denote the empty binary string and $\oplus$ denote concatenation of binary strings
**Require:** model update $u \in \mathbb{R}^d$, quantization step size $\Delta \in \mathbb{R}_{>0}$
**Ensure:** encoded model update $c \in \{0, 1\}^*$
  $q \leftarrow \text{STOCHASTICROUND}(u, \Delta)$
  $c \leftarrow \square;\ i \leftarrow 0$
  **while** $i < d$ **do**
    $r \leftarrow \text{LEADINGZEROS}(q_{i:})$
    $c \leftarrow c \oplus \text{GAMMA}(r + 1);\ i \leftarrow i + r$
    $c \leftarrow c \oplus \text{sign}(q_i) \oplus \text{GAMMA}(|q_i|);\ i \leftarrow i + 1$
  **end while**
  **return** $c$

---

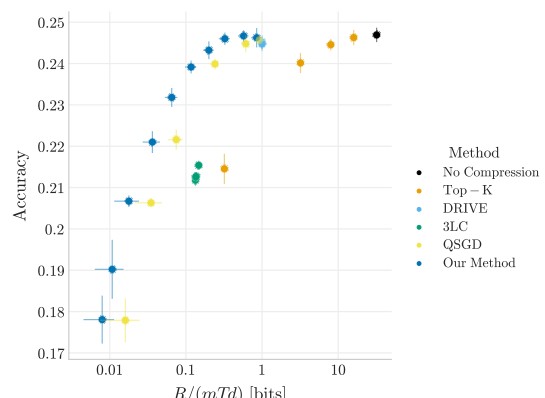

Figure 1: Final model accuracy versus per-coordinate bitrate for a variety of methods.

We use uniform quantization with variable-length bit representations to represent likelier values with shorter bit sequences, optimizing average representation length. STOCHASTICROUND (Forsythe, 1950) randomly quantizes model updates coordinate-wise to integers with a quantization step size $\Delta \in \mathbb{R}_{>0}$. Entropy coding yields binary strings losslessly: we use the GAMMA universal code (Elias, 1975) to encode each nonzero coordinate, as well as the run length of zeros (LEADINGZEROS) preceding each nonzero coordinate. $\mathcal{D}$ simply parses the GAMMA code from the binary string, inserts zeros, recovers signs, and dequantizes to multiples of $\Delta$.

**Rate–Distortion Formulation** We wish to optimize the performance of the model (e.g., final accuracy), such that the total rate $R$ is below an acceptable bitrate budget $B$: $\min f(\theta)$ s.t. $R(\mathcal{Q}) = \sum_{t,k} |\mathcal{E}(u_t^k, \Delta_t^k)| \leq B$, where $|\cdot|$ denotes bit sequence length and $\Delta_t^k$ is the quantization step size used by client $k$ in round $t$, chosen by a policy $\mathcal{Q}$. We consider total distortion as a proxy to model accuracy: $D(\mathcal{Q}) = \sum_{t,k} \|u_t^k - \mathcal{D}(\mathcal{E}(u_t^k, \Delta_t^k), \Delta_t^k)\|_2^2$ (A.3). The global rate–distortion optimum may be found by minimizing the Lagrangian $L(\mathcal{Q}, \lambda) = R(\mathcal{Q}) + \lambda D(\mathcal{Q})$, where $\lambda$ is the Lagrange multiplier controlling the trade-off between $R$ and $D$. Since the Lagrangian is separable across updates, the optimal policy yielding quantization step sizes $\Delta_t^k = \mathcal{Q}(\lambda, u_t^k)$ can be found in a distributed way: each client solves a local problem $\Delta_t^k = \arg\min_\delta |\mathcal{E}(u_t^k, \delta)| + \lambda \|u_t^k - \mathcal{D}(\mathcal{E}(u_t^k, \delta), \delta)\|_2^2$. We find that, empirically, $\Delta_t^k$ is largely independent of $u_t^k$, and there is a monotonic relationship between $\Delta_t^k$ and $\lambda$ (A.3). Thus, we can set $\mathcal{Q}(\lambda, u_t^k) \equiv \Delta$, and let the server control a global $\Delta$ rather than $\lambda$, justifying a simple algorithm that does not require clients to solve for the optimal step size.

## 3 EMPIRICAL RESULTS & DISCUSSION

In experiments (results shown for Stack Overflow NWP, see A.2 for details and A.4 for results across tasks) our method largely outperforms existing approaches in the accuracy–rate trade-off (Figure 1). DRIVE (Vargaftik et al., 2021) and QSGD (Alistarh et al., 2017) are most competitive with our method. The key distinguishing feature of DRIVE is its use of random rotations. For QSGD the key differentiating factor is normalization, which leads to a different effective $\Delta$ per update.

**Ablation: Random Rotations.** Applying a random rotation to the input distribution before quantization will "Gaussianize" the distribution. This can hide the existing structure of the model updates; it increases the per-coordinate entropy, and hence the expected bitrate (recall that a Gaussian is the max-entropy distribution for a given variance). Figure 2 shows that applying a random Hadamard or discrete Fourier transform (DFT) before quantization results in a worse entropy–distortion frontier.

**Ablation: Normalization.** Scaling each model update to the same vector magnitude $\|u_t^k\|$ before quantization is equivalent to choosing a magnitude-dependent per-client $\Delta_t^k$. This leads to an inferior global rate–distortion trade-off. In FL the update magnitudes can vary dramatically (Li et al., 2020). Comparing the $R$–$D$ curves for our method and QSGD, we find that our method provides a better rate–distortion performance, particularly on tasks with significant client heterogeneity (Figure 3).

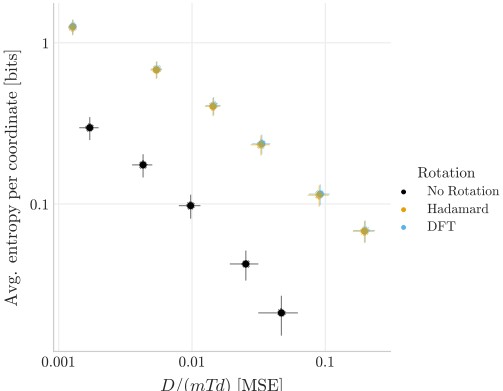

Figure 2: Average per-coordinate entropy versus distortion of model updates after a random rotation (or no rotation) is applied.

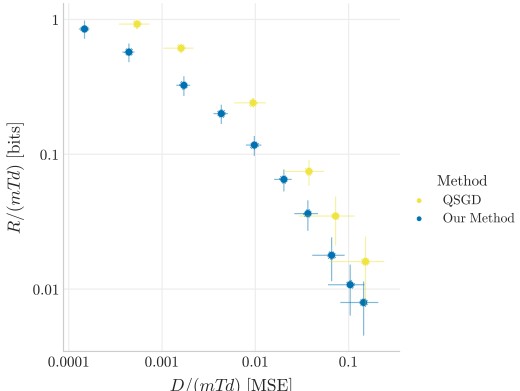

Figure 3: Average per-coordinate rate versus distortion of model updates for QSGD and our method.

URM STATEMENT

The authors acknowledge that at least one key author of this work meets the URM criteria of ICLR 2023 Tiny Papers Track.

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

# A    APPENDIX

## A.1    FEDERATED LEARNING WITH COMPRESSION: A PRIMER

In FL, we often wish to find a model $\theta \in \mathbb{R}^d$ that minimizes a weighted average of client losses

$$\min_\theta f(\theta), \text{ with } f(\theta) = \sum_{k=1}^{K} w_k f_k(\theta) \tag{1}$$

where $K$ is the total number of clients and $f_k$, $w_k$ are the loss function and weight of client $k$. For practical reasons, $w_k$ is often the number of examples held by client $k$ (*example weighting*), which can incur optimization benefits (Li et al., 2020). We wish to solve equation 1 without sharing data and with minimal client-to-server communication. To do so we combine FedOpt (Reddi et al., 2021) (generalizing FedAvg (McMahan et al., 2017)) with compression.

In each round $t$ of FedOpt, the server broadcasts its model $\theta_t$ to a set of clients $\mathcal{S}_t$. Each client $k \in \mathcal{S}_t$ uses LOCALTRAIN to train its model locally. LOCALTRAIN$(\theta, f)$ is often multiple steps of SGD on $f$ starting at $\theta$. After computing $\theta_t^k = $ LOCALTRAIN$(\theta_t, f_k)$, the client sends its weighted update $u_t^k := w_k(\theta_t^k - \theta_t)$ to the server.

To reduce communication, clients can instead send a compressed update $c_t^k := \mathcal{E}(u_t^k)$ to the server, where $\mathcal{E}$ is some encoder. The server decodes the model updates using a decoder $\mathcal{D}$, and computes a weighted average $g_t$ of the $\mathcal{D}(c_t^k)$ (using the weight $w_k$). Finally, the server updates its model using a procedure SERVERUPDATE. As proposed by Reddi et al. (2021), SERVERUPDATE is typically a first-order optimization step, treating $g_t$ as a gradient estimate, with server learning rate $\eta_s$. For example, if SERVERUPDATE is gradient descent, then SERVERUPDATE$(\theta, g) = \theta - \eta_s g$.

Algorithm 2 summarizes our framework. Similar algorithms appear elsewhere (e.g., Haddadpour et al., 2021). The concern of this paper is to develop appropriate encoding and decoding operators $\mathcal{E}, \mathcal{D}$ and control them in a way that is aligned with the global rate–distortion trade-off.

---

**Algorithm 2** FedOpt with compression

---

**Input:** Number of rounds $T$, initial model $\theta_0 \in \mathbb{R}^d$, LOCALTRAIN, SERVERUPDATE, encoder $\mathcal{E}$, decoder $\mathcal{D}$
**for** $t = 0, \ldots, T$ **do**
    $\mathcal{S}_t \leftarrow$ (random set of $m$ clients)
    Broadcast $\theta_t$ to all clients $k \in \mathcal{S}_t$
    **for** each client $k \in \mathcal{S}_t$ **in parallel do**
        $\theta_t^k \leftarrow$ LOCALTRAIN$(\theta_t, f_k)$
        Compute $u_t^k = w_k(\theta_t^k - \theta_t)$
        Send $c_t^k = \mathcal{E}(u_t^k)$ to the server
    **end for**
    $g_t \leftarrow \dfrac{\sum_{k \in \mathcal{S}_t} \mathcal{D}(c_t^k)}{\sum_{k \in \mathcal{S}_t} w_k}$
    $\theta_{t+1} \leftarrow$ SERVERUPDATE$(\theta_t, g_t)$
**end for**

---

## A.2    EXPERIMENTAL SETUP

In order to inform the design of our method, we perform empirical evaluations of Algorithm 2 on a variety of federated tasks drawn from benchmarks in (Reddi et al., 2021). For the body of this paper we show performance on the Stack Overflow next-word prediction task. We observe similar trends on other tasks, and include results in A.4.

**Datasets, Models, and Tasks.**    We use three datasets: CIFAR-100 (Krizhevsky, 2009), EM-NIST (Cohen et al., 2017), and Stack Overflow (Authors, 2019). For CIFAR-100, we use the client partition proposed by Reddi et al. (2021). The other two datasets have natural client partitions where each client is an author (of handwritten digits and forum posts). For CIFAR-100, we train a ResNet-18, replacing batch normalization with group normalization (see (Hsieh et al., 2020)). For EMNIST,

we train a network with two convolutional layers, max-pooling, dropout, and two dense layers. For Stack Overflow, we perform next-word prediction (NWP) using an RNN with a single LSTM layer, and tag prediction (TP) using a multi-class logistic regression model. For a summary of the dataset statistics, tasks, and models used, see Table 1.

Table 1: Datasets, Tasks & Models

| DATASET | NUM CLIENTS | | NUM EXAMPLES | | TASK | MODEL |
|---------|-------|------|-------|------|------|-------|
| | TRAIN | TEST | TRAIN | TEST | | |
| EMNIST | 3,400 | 3,400 | 671,585 | 77,483 | CHARACTER RECOGNITION | CNN |
| STACK OVERFLOW | 342,477 | 204,088 | 135.8M | 16.6M | NEXT-WORD PREDICTION | LSTM |
| STACK OVERFLOW | 342,477 | 204,088 | 135.8M | 16.6M | TAG PREDICTION | LOGISTIC REGRESSION |
| CIFAR-100 | 500 | 100 | 50,000 | 10,000 | IMAGE RECOGNITION | RESNET-18 WITH GROUPNORM |

**Algorithms.** We focus on two special cases of Algorithm 2: FedAvg (McMahan et al., 2017) and FedAdam (Reddi et al., 2021). In both, LOCALTRAIN is $E$ epochs of mini-batch SGD with client learning rate $\eta_c$. For FedAvg and FedAdam, SERVERUPDATE is SGD or Adam (respectively) with server learning rate $\eta_s$. We use FedAvg and FedAdam on the vision tasks (CIFAR-100 and EMNIST), but only FedAdam on the language tasks (Stack Overflow), as FedAvg performs poorly there (Reddi et al., 2021). We set $E = 1$ and use a batch size of 32 throughout. We perform $T = 1500$ rounds of training for each task. At each round, we sample $m = 50$ clients uniformly at random. We tune $\eta_c, \eta_s$ over $\{10^{-3}, 10^{-2}, \ldots, 10\}$ by selecting the values that minimize the average validation loss over 5 random trials.

**Other Benchmarks.** We evaluate our compression method against existing approaches on the tasks described above (results shown on Stack Overflow NWP in Figure 1). As a baseline, we include runs with NOCOMPRESSION, where $(\mathcal{E}, \mathcal{D})$ are no-ops and clients communicate their weighted updates at full-precision. This method has fixed rate at 32-bits per coordinate and no distortion. The accuracy achieved with NOCOMPRESSION can be understood as the target accuracy we aim to reach with compression. We also compare to TOP-K (Aji & Heafield, 2017), DRIVE (Vargaftik et al., 2021), 3LC (Lim et al., 2019), and QSGD (Alistarh et al., 2017).

**Computational Tractability.** Given that we designed this method to be practical for use in real FL scenarios, we took care to ensure it is computationally tractable. Analyzing compression speed in a simulated environment across tasks and $\Delta$, we find on average less than 3% of training time is spent on encode, stochastic round and decode ops. The compression percentage of training time is dependent on model architecture, as encoding time scales with model parameters while training time scales with model depth. Compression time can be reduced with system and hardware optimizations.

## A.3 EMPIRICAL JUSTIFICATION

**Statistical Structure of Model Updates.** We observe a relatively consistent statistical structure across all tasks: client updates tend to resemble a symmetric power law distribution with a spike at zero (Figure 4).

**Distortion as a Proxy for Model Accuracy.** We verify empirically that for varying $\Delta$, total distortion is a good proxy for model performance (Figure 5). This relationship holds across tasks.

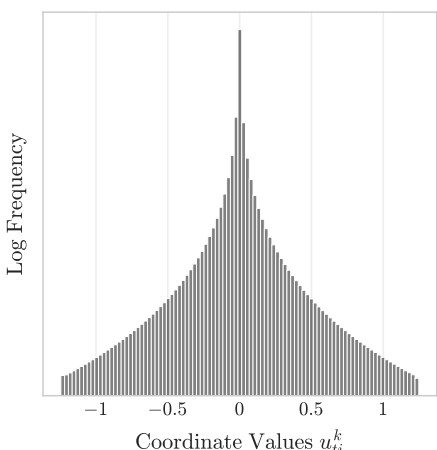

Figure 4: Histogram of coordinate values of weighted client updates, averaged over the course of training across all participating clients in Stack Overflow NWP.

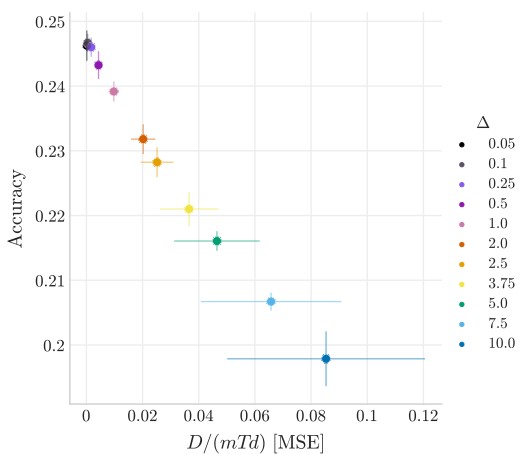

Figure 5: Accuracy of a model versus average per-coordinate distortion for various quantization step sizes $\Delta$ on Stack Overflow NWP. Error bars indicate the variance over 5 random trials.

**Agreement in Optimal $\Delta$ Across Clients.** For varying $\lambda$, we let clients solve $\Delta_t^k = \arg\min_\delta |\mathcal{E}(u_t^k, \delta)| + \lambda \lVert u_t^k - \mathcal{D}(\mathcal{E}(u_t^k, \delta), \delta) \rVert_2^2$ by grid search, selecting from a pre-determined set of $\delta$'s. We observe significant agreement on $\Delta_t^k$ across clients and rounds for any given $\lambda$ (Figure 6). Meaning that to minimize their local rate–distortion objective specified by the given $\lambda$, clients select a consistent quantization step size $\Delta$. We find that this relationship holds across different architectures, tasks, and optimizers.

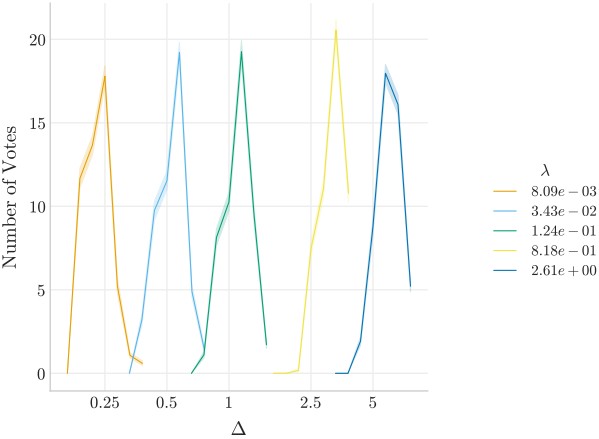

Figure 6: Histogram of client-selected quantization step sizes $\Delta_t^k$ for a given $\lambda$, averaged across training rounds, on Stack Overflow NWP.

### A.4 FULL RESULTS ACROSS TASKS

**Overall Performance.** Our method significantly outperforms Top-K, DRIVE, and 3LC. Performance is similar to QSGD on tasks with relatively similar data on clients (CIFAR-100 and EMNIST), with a visible gap in Stack Overflow tasks.

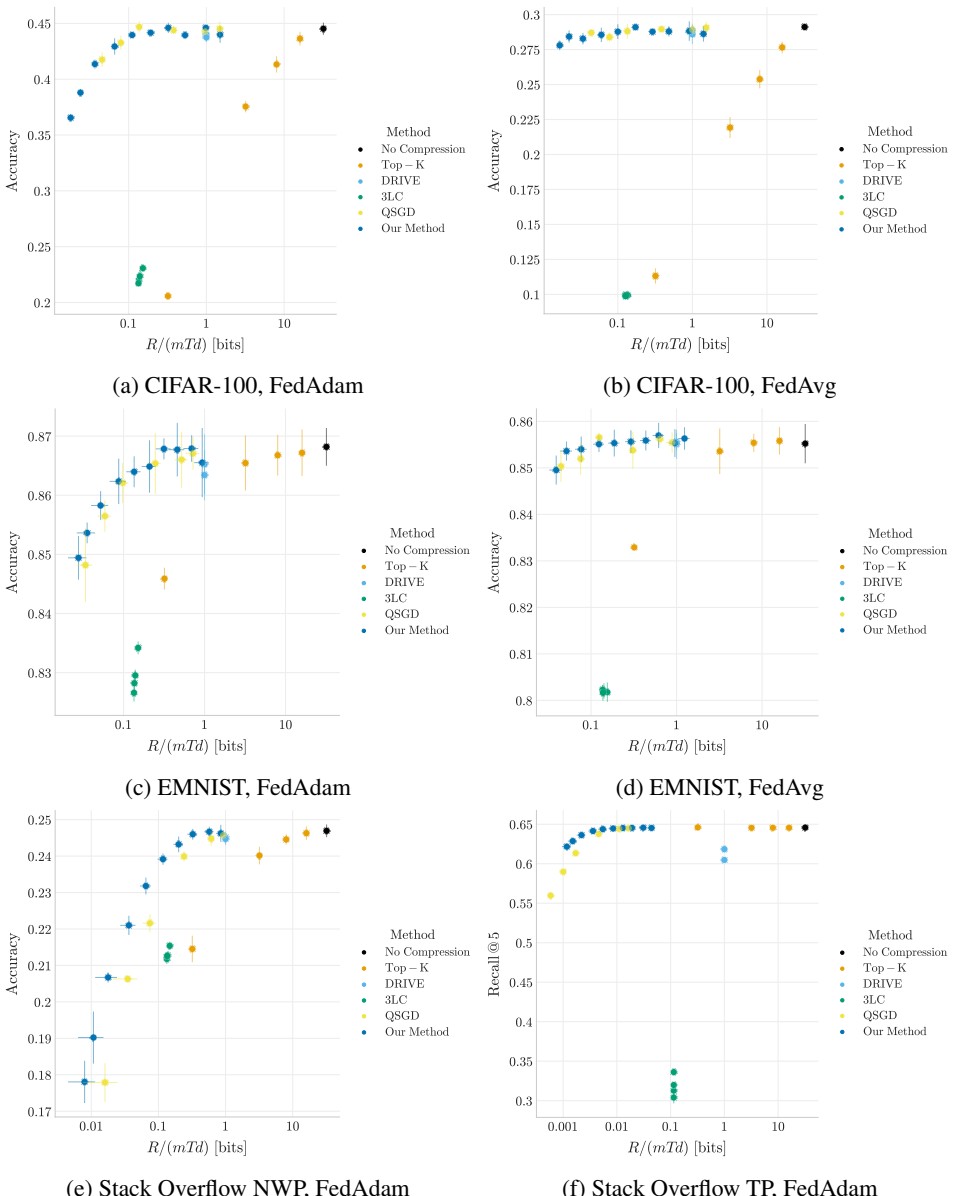

Figure 7: Our method performs competitively against all others in terms of the accuracy–communication cost trade-off. We use a range of $\Delta \in \{0.05..17.5\}$. Error bars indicate variance in average per-coordinate rate and final model accuracy over five random trials.

**Rotation Ablation.** The rate–distortion tradeoff is significantly better without rotation for EMNIST and Stack Overflow datasets. It is only slightly better for CIFAR-100, due to the fact that the client updates are not as sparse as for the other tasks.

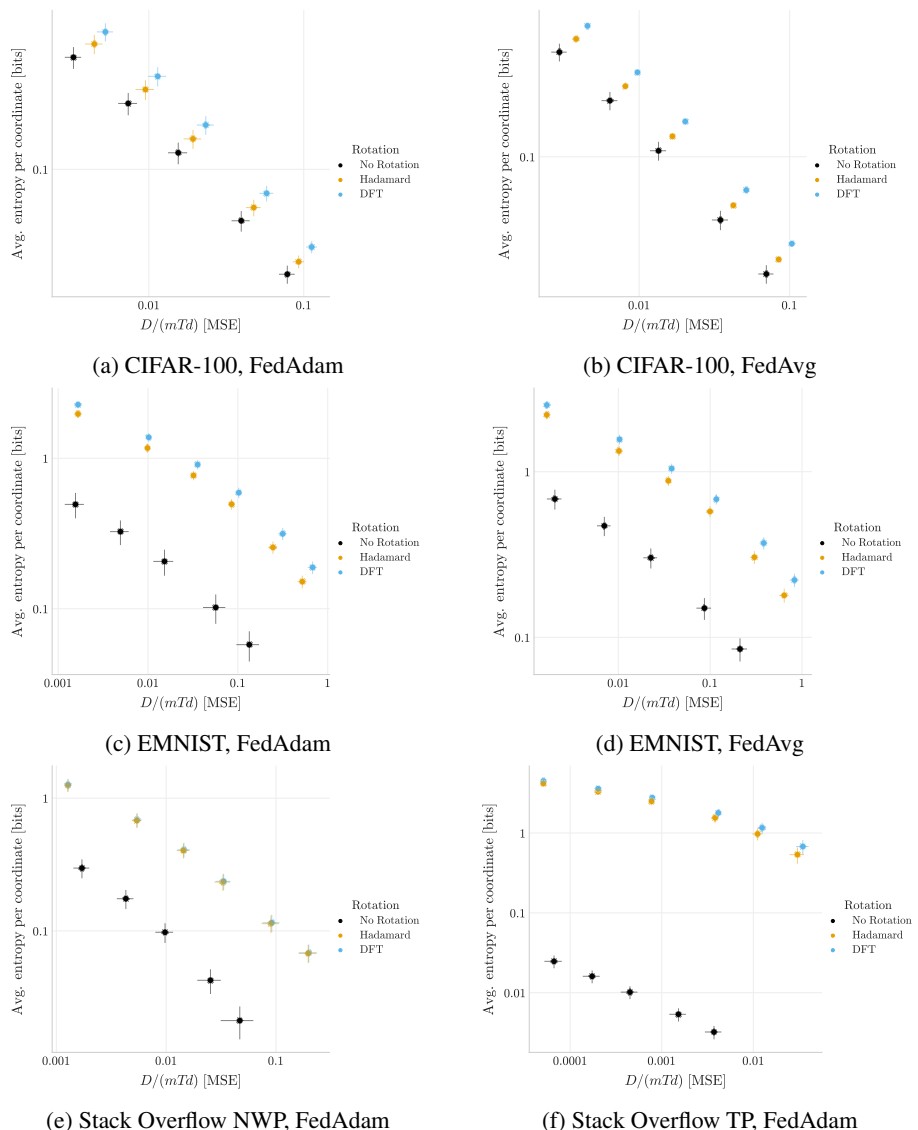

(a) CIFAR-100, FedAdam

(b) CIFAR-100, FedAvg

(c) EMNIST, FedAdam

(d) EMNIST, FedAvg

(e) Stack Overflow NWP, FedAdam

(f) Stack Overflow TP, FedAdam

Figure 8: Transforming the client updates via a random rotation increases both entropy and distortion, producing a distribution with a worse entropy–distortion frontier. The effect is most dramatic on highly structured client updates. Error bars indicate variance in average per-coordinate distortion and average per-coordinate entropy over five random trials.

**Normalization Ablation.** QSGD yields similar performance on the CIFAR-100 dataset, where the clients have the same amount of training data. The gain in performance by our method is more pronounced on the more heterogeneous Stack Overflow tasks.

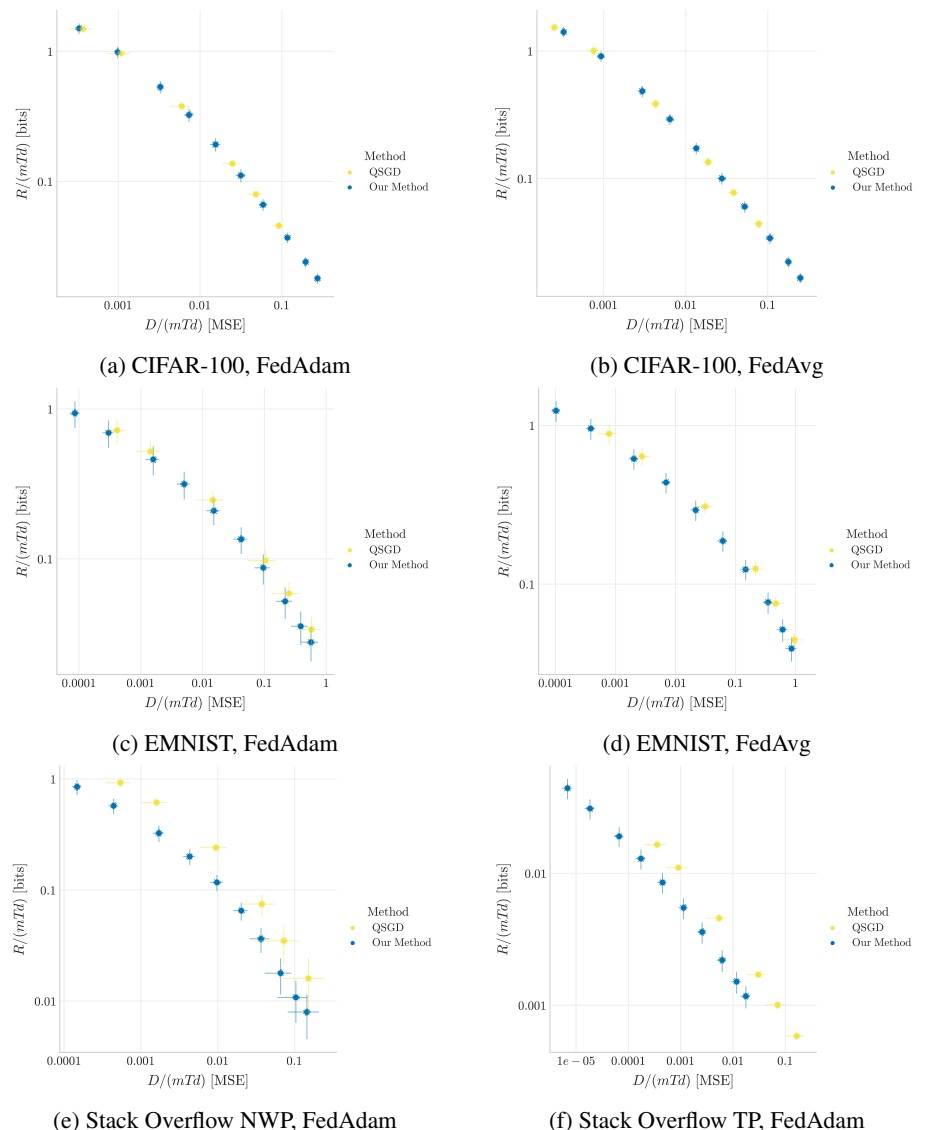

Figure 9: Transforming heterogeneous client updates via normalization results in a slightly worse rate–distortion performance. The effect is negligible on tasks with client updates of the same magnitude. Error bars indicate variance in average per-coordinate distortion and average per-coordinate rate over five random trials.

