# OpenReview forum: "A Rate--Distortion View on Model Updates"
_ICLR.cc/2023/TinyPapers — Submitted to Tiny Papers @ ICLR 2023_

### Official Review · Reviewer_z1sw · 2023-03-27

**Confidence:** 3

**Summary Of Contributions:**

Rate-distortion theory was utilized to present a compression method. This method is claimed to reduce communication costs in Federated Learning. An algorithm is presented to design the encoder operator for clients.

**Rating:**

Clear, Correct, and Reproducible (CCR): a submission which meets the reviewing criteria

**Strengths And Weaknesses:**

Strengths :

* The paper addresses communication efficiency which is an important component in FL
* Empirical results were provided for variety of datasets along different modalities


Weaknesses :

* It appears the proposed algorithm addresses average case.
* Emperical results on large scale datasets are not available.

**Suggested Changes:**

The authors can also extend and produce results for large scale datasets and settings. This can definitely enhance the impact of this work.

---

### Official Review · Reviewer_m7wr · 2023-03-28

**Confidence:** 3

**Summary Of Contributions:**

This paper proposes a new compression method using knowledge of rate–distortion theory. Several experiments were conducted to show how the proposed method outperforms several state-of-the-art methods in the accuracy-rate trade-off.

**Rating:**

High Potential (HP): a submission which meets the reviewing criteria and has potential to make an impact on the field

**Strengths And Weaknesses:**

S: A new compression method for FL that (empirically) outperforms the state-of-the-art methods. Ablation studies also strengthen the paper.
W: Missing formalization but this is explicit in the paper, which can also be investigated in future work.

**Suggested Changes:**

- The acronym FL in the abstract is not used within the abstract and can be removed;
- What is d in Algorithm 1? Please define it.

---

### Author Response · Authors · 2023-06-01
**Response to the reviewers**

We would like to thank the reviewers for their comments and suggestions.

To reviewer m7wr:
- We removed the unused FL acronym in the abstract.
- $d$ is the dimensionality of the client update, defined as $u \in R^d$.

To reviewer z1sw:
- Due to time constraints, we were not able to run experiments on larger-scale datasets, though we agree such an exploration would strengthen the impact of this work.

 We made a revision that incorporates the discussed changes.

---

### Comment · Area_Chair_k39o · 2023-06-06
**Archival Criterion Check**

This work meets the threshold for archival, contents the URM statement and is deanonymized.

---

### Meta-Review · Area_Chair_k39o · 2023-04-08

**Recommendation:** Invite to present
**Confidence:** 3

**Metareview:**

The authors proposed a novel compression method for federated learning to enhance communication efficiency. The model is well-defined, and the results are promising. Overall, the presentation of the paper is of high quality.

The pros and cons summarized from reviewer's comments:
Pros:

1. The paper introduces a new compression method for Federated Learning that outperforms state-of-the-art methods.
2. The ablation studies makes the paper more compelling.
3. The focus on communication efficiency is closely related to gradient compression for distributed training, which is an important part for federated learning.
4. Empirical results are provided for a variety of datasets across different modalities.

Cons:

1. The paper lacks formalization, which could be investigated in future work.
2. The proposed algorithm addresses the average case.
3. Empirical results on large-scale datasets are not available.


**Summary:**

The paper presents a rate-distortion theory-based compression method for Federated Learning, demonstrating improved accuracy-rate trade-offs and providing an algorithm for designing client encoder operators.

**Comments And Feedback To The Authors:**

You proposed a novel compression method for federated learning to enhance communication efficiency. The model is well-defined, and the results are promising. Overall, the presentation of the paper is of high quality.

**Reason For Not Giving A Higher Recommendation:**

As mentioned by reviewer z1sw, the paper omits empirical results on large-scale datasets. Including these results would make the paper more compelling.

**Reason For Not Giving A Lower Recommendation:**

N/A

---

### Decision · Program_Chairs · 2023-04-08

Invite to present